# Neighborhood context, genetic influences, and life satisfaction: Evidence from the German twin family panel

**Nadia V. Harerimana**[1,2☯]*, **Yixuan Liu**[2☯], **Mirko Ruks**[2]

1 Max Planck Research Group Biosocial – Biology, Social Disparities, and Development, Max Planck Institute for Human Development, Berlin, Germany, 2 Faculty of Sociology, Bielefeld University, Bielefeld, Germany

☯ These authors contributed equally to this work.
* nadia.victoria.harerimana@gmail.com

## Abstract

Both genetic influences and neighborhood environments play a role in shaping life satisfaction. However, research examining gene-environment interactions (GxE) in this context remains limited. This study investigates how neighborhood deprivation moderates the effects of genetic influences on life satisfaction. Using data from 760 dizygotic (DZ) twin pairs in the German Twin Family Panel (TwinLife), we apply twin fixed-effect models to estimate GxE effects. Results indicate that a Polygenic index (PGI) for subjective well-being is positively associated with life satisfaction. Notably, this association is strongest among individuals living in moderately deprived neighborhoods, and weaker in both highly deprived and less deprived areas. Thus, there are signs of compensation in less deprived areas and, particularly, diathesis-stress/triggering in highly deprived areas.

## 1. Introduction

Life satisfaction, a fundamental element of subjective well-being, encapsulates how individuals cognitively evaluate their lives. Beyond its intrinsic value, research highlights the multifaceted benefits associated with life satisfaction. Studies indicate its correlation with various positive life outcomes, including longevity, enhanced health, academic achievement, income levels, and involvement in civic activities [1–12]. Given these broad implications, a comprehensive understanding of the factors that influence life satisfaction is of significant theoretical and practical importance.

Genetic factors represent one significant contributor to individual differences in life satisfaction. Findings from behavioral genetics consistently indicate that genetic variation is associated with variability in life satisfaction across individuals. Meta-analyses report heritability estimates ranging from 32% [13] to 40% [14] suggesting that a substantial proportion of the variance in life satisfaction can be attributed to

**Data availability statement:** The TwinLife data are archived at GESIS, Cologne: ZA6701 Data File Version 8.0.0, https://doi.org/10.4232/1.14331, and are available for free (after application) in the GESIS data catalog for scientists. The genetic data are available to collaborators through TwinLife data management email at twinlife@uni-bielefeld.de.

**Funding:** This work was made possible by the participants of the German Twin Family Panel (TwinLife). The TwinLife study was funded by the German Research Foundation (Grant No. 220286500), with funding awarded to Martin Diewald, Christian Kandler, Frank M. Spinath, Bastian Mönkediek, and Rainer Riemann. The molecular genetic extension of TwinLife was also funded by the German Research Foundation (Grant No. 428902522), with funding awarded to Martin Diewald, Peter Krawitz, Markus M. Nöthen, Rainer Riemann, and Frank M. Spinath. The funders had no role in the study design, data collection and analysis, decision to publish, or manuscript preparation. MR is supported by the TwinLife project, YL is supported by the TwinSNPs and TECS projects,while NVH receives support from the European Union's HORIZONMSCA-2021-DN-01 programme (Grant No. 101073237).

**Competing interests:** The authors declare that they have no known competing financial interests or personal relationships that could have appeared to influence the work reported in this manuscript titled "Neighborhood Deprivation, Genetic Influences, and Life Satisfaction - Evidence from the German Twin Family Panel." The research was conducted independently, and the funding sources were not involved in the study design, data collection, analysis, interpretation of data, writing of the report, or the decision to submit the article for publication. All authors have approved the final manuscript and have agreed to its submission to PLOS One Journal.

genetic influences [15,16]. In parallel, environmental factors also play a crucial role in shaping life satisfaction, with neighborhood characteristics receiving considerable empirical attention. Research has shown that residing in disadvantaged neighborhoods increases the risk of mortality and mental illnesses such as depression and anxiety [17–19]. This association may stem from various factors including poor physical conditions, limited access to resources, exposure to environmental toxins, and a lack of social cohesion and safety [20–23]. Unfavorable neighborhood conditions have been linked to lower life satisfaction and overall poorer health outcomes among residents [24,25] and individuals in better-off neighborhoods generally experiencing better health outcomes [26–28].

Both genetic and neighborhoods are important for life satisfaction. In this regard, the research on geneenvironment interactions (GxE) investigates how the genetic influences vary as a function of the environmental context and vice versa [29]. In fact, multiple studies have investigated whether genetic effects vary as a function of neighborhood deprivation. Most of these studies focus on health or psychopathological outcomes or overall problem behavior. A twin study by Singh et al. [30] reports that perceived neighborhood cohesion moderates the genetic effect on internalization but not on externalizing behavior. For females, the relevance of genetics decreases with higher neighborhood cohesion, they find an opposite pattern for males. Strachan et al. [31] found that genetic influences on depression increases with increasing neighborhood deprivation. In line with that, Carroll et al. [32] find that genetic influences on ADHD are higher among individuals living in disadvantaged neighborhoods. Rhew et al. [33] report that while neighborhood deprivation moderates environmental influences on hazardous drinking, it does not moderate genetic influences. Dash et al. [34] show that genetic influences on externalizing problem behavior are weaker for individuals living in neighborhoods with lower educational, or social and economic opportunities. Burt et al. [35] show that genetic influences on antisocial behavior in childhood are greater among individuals living in advantaged neighborhoods.

Importantly, none of the previous studies have specifically focused on life satisfaction as the primary outcome in G × E research. However, a meta-analysis by Røysamb et al. [16] highlights significant heterogeneity in heritability estimates for life satisfaction, which may indicate the presence of G × E effects. Despite this, research on G × E in the context of life satisfaction remains surprisingly limited compared to other behavioral traits [13]. Only a few studies have examined whether the heritability of life satisfaction varies across environmental contexts, such as family socioeconomic status [36], marital status [37], or parental divorce [38].

Therefore, this study aims to address a critical gap in the literature by examining whether the genetic influence on life satisfaction varies as a function of neighborhood deprivation. To investigate G × E interactions, the study employs a co-twin control design, which leverages the natural experiment of sibling differences to account for potential confounding effects and overcome prior methodological limitations. This research explores how contextual factors at the neighborhood level moderate genetic influences on life satisfaction by integrating geo-coded data on neighborhood deprivation with individual- and household-level survey data from the longitudinal

TwinLife study of German twins and their immediate relatives. As such, it represents a pioneering effort to uncover the interplay between genetic and environmental conditions at the neighborhood level in influencing individual life satisfaction in Germany.

## 2. Theoretical framework

In this section, we discuss theoretical arguments on how and why the genetic effect on life satisfaction may vary as a function of neighborhood deprivation. To do so, in a first step we distinguish general GxE mechanisms. Then, we link these GxE mechanisms to the mechanisms of neighborhood effects discussed in the literature to explain how neighborhood deprivation might affect the genetic effect on life satisfaction.

### 2.1. GxE mechanisms

Shanahan and Hofer [29] distinguish four GxE mechanisms. Triggering describes situations where exposure to disadvantaged environmental contexts, characterized by high levels of risk and stressors, increases the probability of realizing genetic vulnerabilities. Compensation refers to cases where the realization of a genetic risk is limited in advantaged environments. Social control is a specific form of compensation that focuses on the enforcement of social norms and values in advantaged environments restricting the realization a genetic risk. Finally, enhancement refers to a multiplication process where the realization of a positive genetic correlate is amplified in advantaged environments.

It's possible to categorize these mechanisms based on the type of genetic disposition and environmental context. However, Shanahan and Hofer [29] don't delve into how a disadvantaged environment can affect the realization of a positive genetic potential. To address this, we introduce the concept of suppression, where the influence of a genetic association is weakened under adverse environmental conditions. Overall, while triggering and compensation imply stronger genetic effects in disadvantaged environments (negative GxE), suppression and enhancement explain stronger genetic effects in advantaged environments (positive GxE).

An overview of these GxE mechanisms is provided in Table 1.

### 2.2. GxNeighborhood mechanisms

Various theoretical mechanisms driving neighborhood effects have been discussed in the literature [for an overview, see, e.g., [39,40–43]. Here, we first discuss five mechanisms underlying the effects of neighborhood deprivation. Then we link these mechanisms to the overall GxE typology derived before to explore how neighborhood deprivation might moderate the genetic effect on life satisfaction.

First, epidemic models emphasize the role of peers within the neighborhood for socialization. As human behavior is shaped partly by social learning [44], it is assumed that neighborhoods can impact individual behavior through interaction with peers in the area. In this view, individuals tend to mirror the behavior of their peers in terms of a contagion effect ("like begets like") [41]. In disadvantaged neighborhoods, peers may exhibit higher levels of problem behavior [45], potentially leading to negative outcomes such as increased victimization [46] or depressive symptoms [47]. Conversely,

**Table 1. Expected GxE mechanisms.**

| | | Environment | Expected GxE pattern |
|---|---|---|---|
| | | Low ("Risk") | High ("Potential") |
| Genetic Disposition | Low ("Risk") | Triggering | Compensation/Social Negative Control |
| | High ("Potential") | Suppression | EnhancementPositive |

in advantaged neighborhoods, individuals are more likely to encounter successful and prosocial peers, fostering social support and opportunities for positive friendships [48–50].

Second, collective socialization models highlight the influence of non-parent adults in a neighborhood on children and adolescents [41]. These adults can serve as role models, shaping aspirations and defining what behavior is considered acceptable [39,41]. In disadvantaged neighborhoods, exposure to negative role models may be more prevalent, promoting attitudes that devalue education and encourage problematic behavior. Additionally, limited job opportunities for adults in these areas may lead adolescents to perceive little benefit from responsible behavior, resulting in lower commitment to education [39,50]. Conversely, in affluent neighborhoods, positive adult role models can inspire, guide, and support individuals, fostering higher aspirations, self-efficacy, and motivation. These adults may also serve as enforcers, monitoring behavior, maintaining order, and intervening when necessary.

A third mechanism operates through the institutional resources available in a neighborhood, which can significantly impact adolescent development [51,52]. These resources, such as schools, libraries, and social services, vary in quality, quantity, and diversity depending on the neighborhood's status. In affluent neighborhoods, schools often have better facilities and teachers, and there is a wider range of social and cultural activities and extracurricular opportunities, such as sports, music, or art. Conversely, in disadvantaged neighborhoods, these institutional resources may be lacking, placing adolescents at greater risk of engaging in problematic or delinquent behavior and limiting their ability to develop talents and skills essential for future socioeconomic success [39].

The fourth mechanism highlights the significance of social capital, which encompasses the connections and networks among individuals, along with the norms of reciprocity and trust [53]. Neighborhood social capital encompasses various dimensions, including the level of social ties, frequency of interactions, and neighboring patterns [54–56]. These social ties within a neighborhood can be advantageous, fostering a cohesive community where residents support each other and share valuable information and resources. In affluent neighborhoods, there tends to be a higher quantity and quality of social ties, along with shared resources, providing residents with a stronger sense of community and access to valuable knowledge about status attainment and job opportunities. Conversely, in deprived neighborhoods, social ties and shared resources may be lacking or significantly reduced.

Lastly, collective efficacy refers to a neighborhood's ability to mobilize informal social control, organize, and pursue common goals, relying on mutual trust, shared norms, and expectations [57]. Neighborhoods with clear rules and trust among residents are more likely to successfully activate social control, leading to positive outcomes such as reduced violence, disorder, and better access to public services. Additionally, residents may feel more empowered and engaged in local events, fostering a sense of belonging. Studies have shown that higher collective efficacy is associated with reduced criminal victimization, deviant behavior, and better mental health [57–63].

These mechanisms can be related to the overall GxE mechanisms discussed before, explaining why neighborhood deprivation might moderate the genetic effect on life satisfaction. On the one hand, in affluent neighborhoods the exposure to 1) positive peer influences, 2) motivating and involved role models, 3) the high quality and availability of supporting institutions, 4) higher quality and quantity of social capital, and 5) higher levels of collective efficacy may have a compensating function so that individuals develop high levels of life satisfaction despite low genetic endowment for life satisfaction. In line with that, in deprived neighborhoods, the lack of these experiences may act as a stressor, triggering the manifestation of genetic disadvantages in terms of low life satisfaction. As a result, we would expect the genetic effect size to increase with higher levels of neighborhood deprivation (H1). On the other hand, the positive neighborhood experiences in affluent neighborhoods may actually enhance the realization of a genetic potential for life satisfaction so that individuals in affluent neighborhoods profit disproportionately from a high genetic disposition for life satisfaction. In deprived neighborhoods, in turn, the lack of these experiences may act as a suppressor, preventing the realization of an actually high genetic endowment for life satisfaction. From this, we would expect the genetic effect size to decrease with higher levels of neighborhood deprivation (H2).

## 3. Data and methods

### 3.1. Participants

The German Twin Family Panel (TwinLife), which began in 2014, is a nationally representative cohortsequential panel-study of four cohorts of same-sex twins and their biological families [64,65]. Unlike many convenience twin studies based on self-selection of twins, TwinLife applies a register-based probability design. It has been shown that the Twin-Life includes families from the whole socio-economic spectrum in Germany [66] and the socio-demographic structure is comparable to other German family surveys [67]. In this study, we included participants from Cohorts 2 (twins born in 2003/2004), 3 (twins born in 1997/1998), and 4 (twins born between 1990 and 1993) who participated in both the first face-to-face (F2F1) wave and the third face-to-face (F2F3) wave, and who provided saliva samples during F2F3. The field period for F2F1 was from September 28, 2014, to May 28, 2015 (Subsample A), and from September 16, 2015, to April 18, 2016 (Subsample B). The field period for F2F3 was from November 26, 2018, to July 6, 2019 (Subsample A), and from September 16, 2019, to June 6, 2020 (Subsample B).

### 3.2. Ethical statement

The data analyzed in this study were obtained from participants of the TwinLife project. Genetic data, including saliva samples for DNA extraction, were collected as part of two TwinLife satellite projects: the molecular genetic extension project (TwinSNPs) and the epigenetic change satellite project (TECS). Ethical approval for the TwinLife study was granted by the German Psychological Society (Protocol Number: RR 11.2009). The protocols for genetic sampling via saliva were additionally reviewed and approved by the Ethics Committee of the Medical Faculty at the University of Bonn (Approval Number: 113/18).

### 3.3. Measures

**3.3.1. Life satisfaction.** Life satisfaction was measured via the "Satisfaction with Life Scale" (SWLS) [68] for participants aged

16 years and above as well as using an adapted version for children aged 10–15 years (SWLS-C) [69].

Participants were asked, "How satisfied are you with your life in general?" and responded to a set of five statements reflecting their cognitive evaluation of overall life satisfaction using a five-point Likert scale, ranging from 1 ("Disagree strongly") to 5 ("Agree strongly"). For individuals aged 16 years and older, the items included: (1) "My life is close to ideal," (2) "The conditions of my life are excellent," (3) "I am satisfied with my life," (4) "So far, I have gotten the important things I want in life," and (5) "If I could live my life over, I would change almost nothing." For participants aged 10–15 years, the items were as follows: (1) "My life is going well," (2) "I have a good life," (3) "I am happy with my life," (4) "I have gotten the things I want in life," and (5) "If I could live my life over, I would want the same life." The final life satisfaction score was derived separately for each version through confirmatory factor analysis (CFA) of the raw indicators, allowing for the estimation of a latent construct that captures the underlying structure of life satisfaction. For participants aged 16 years and older, the model fit was: $\chi^2(5) = 17.10$, $\chi = .004$, RMSEA = .054, CFI = .992, TLI = .984, SRMR = .016. The reliability indices were: Cronbach's $\alpha = .84$, and McDonald's $\omega = .85$. For participants aged 10–15 years, the model fit was: $\chi^2(5) = 26.36$, $\chi < .001$, RMSEA = .077, CFI = .988, TLI = .976, SRMR = .018. The reliability indices were: Cronbach's $\alpha = .86$, McDonald's $\omega = .87$.

**3.3.2. Genetic influences of life satisfaction.** The genetic influence of life satisfaction was measured via a Polygenic index (PGI) for subjective well-being. Participants' saliva was extracted in the survey wave F2F3 (2018–2020) [65,66]. PGI for subjective well-being was estimated for TwinLife participants in Wave 1 using results from the well-being spectrum [70] following the PRS-CS approach [71]. Briefly, genetic variants were filtered for (i) an imputation info score >= 0.6 if available in the respective summary statistics, (ii) a minor allele frequency >= 0.01 in the 1000 Genomes phase 3 European LD

reference panel, and (iii) presence in both the respective summary statistics and the imputed genotype data of TwinLife. For a detailed description of sampling, genotyping, quality control, filtering, and PGI calculation, please see Supplement B-I of the study [72]. Subsequently, associations between PGI and neighborhood index were examined, adjusting for sex, 10 genetic principal components using regression modeling.

   **3.3.3. Neighborhood deprivation.** Neighborhood deprivation is measured using the German Index of Socioeconomic Deprivation (GSID) developed by Michalski et al. [73] which is publicly available [74]. The index was created based on administrative data for Germany published by the INKAR database and comprises three dimensions (Education, Employment, and Income) each of which is measured via three indicators. The GSID was extracted via principal component analysis and matched to the TwinLife survey data based on district codes. To account for possible non-linearities, we categorized the GSID score along the terciles, distinguishing low, middle, and high neighborhood deprivation.

   **3.3.4. Covariates.** As families might select into neighborhood deprivation partly as a function of family socio-economic status or age structure, in the statistical analysis we additionally control for family SES, measured as a factor based on parental years of education, occupational status (ISEI) and OECD household net income through CFA. The CFA model for family SES was saturated, yielding perfect fit indices: $\chi^2(0) = 0$, $\chi = $ NA, RMSEA = 0, CFI = 1.00, TLI = 1.00, SRMR = 0. The reliability indices were: Cronbach's $\alpha$ = .40 and McDonald's $ = .70. Although Cronbach's $\alpha$ was relatively low—likely due to violations of the tau-equivalence assumption, McDonald's $\omega$ indicated acceptable construct reliability. Given that $\omega$ does not assume equal factor loadings and is more appropriate in structural equation modeling contexts, we consider the construct to demonstrate adequate internal consistency. Additionally, we control for birth cohort status, using a categorical variable indicating whether a twin belongs to cohort 2, 3, or 4. Finally, Table 2 provides a descriptive overview of the full sample (N = 1544) and Table 3 splits up the descriptives by zygosity (N DZ twins = 760; N MZ twins = 784), showing no meaningful differences across zygosity.

## 3.4. Statistical analysis

To test the competing GxE hypotheses, we apply a step-wise regression approach. In a first step, we estimate a multi-level model accounting for the family structure on the pooled sample. In a second step, we estimate a fixed effects model on the DZ twin sample, accounting for unobserved between-family hetero-geneity. Overall, with twin i clustered in twin pair j, the model equation is given by:

$$Y_{ij} = \beta_1 * \dot{PGS}_{ij} + \beta_2 * GSID_j + \beta_3 * \dot{PGS}_{ij} \times GSID_j + \beta_4 * Z_j + \epsilon_{ij}$$

Here, $PGS_{ij}$ is the measured genotype, $GSID_j$ is the neighborhood deprivation, varying only between twin pairs, and $Z_j$ is a vector of controls. Importantly, in order to get unbiased estimates of the PGS for subjective well-being ($\beta_1$) and of its

**Table 2. Descriptive statistics of the full sample (N = 1544).**

|  | Min | Max | Mean | SD |
|---|---|---|---|---|
| Life Satisfaction | −4.49 | 1.38 | 0.00 | 1.00 |
| PGS | −4.13 | 3.15 | −0.00 | 1.00 |
| Deprivation Index | −2.51 | 2.30 | −0.00 | 1.00 |
| SES | −2.41 | 2.51 | 0.00 | 1.00 |
| Cohort 2 (10–12) | 0.00 | 1.00 | 0.46 | 0.50 |
| Cohort 3 (16–18) | 0.00 | 1.00 | 0.36 | 0.48 |
| Cohort 4 (22–25) | 0.00 | 1.00 | 0.18 | 0.38 |
| Female | 0.00 | 1.00 | 0.56 | 0.50 |
| Age | 10.00 | 25.00 | 15.31 | 4.52 |

**Table 3. Descriptive statistics by zygosity (N DZ twins = 760; N MZ twins = 784).**

| | DZ | | | | MZ | | | |
|---|---|---|---|---|---|---|---|---|
| | Min | Max | Mean | SD | Min | Max | Mean | SD |
| Life Satisfaction | −4.14 | 1.38 | −0.03 | 1.00 | −4.49 | 1.38 | 0.03 | 0.99 |
| PGS | −3.03 | 3.15 | 0.05 | 0.99 | −4.13 | 3.02 | −0.05 | 1.00 |
| Deprivation Index | −2.51 | 2.26 | 0.00 | 0.95 | −2.51 | 2.30 | −0.01 | 1.05 |
| SES | −2.40 | 2.33 | 0.07 | 0.99 | −2.41 | 2.51 | −0.07 | 1.00 |
| Cohort 2 (10–12) | 0.00 | 1.00 | 0.53 | 0.50 | 0.00 | 1.00 | 0.40 | 0.49 |
| Cohort 3 (16–18) | 0.00 | 1.00 | 0.37 | 0.48 | 0.00 | 1.00 | 0.35 | 0.48 |
| Cohort 4 (22–25) | 0.00 | 1.00 | 0.11 | 0.31 | 0.00 | 1.00 | 0.25 | 0.43 |
| Female | 0.00 | 1.00 | 0.54 | 0.50 | 0.00 | 1.00 | 0.57 | 0.49 |
| Age | 10.00 | 24.00 | 14.46 | 4.03 | 10.00 | 25.00 | 16.13 | 4.82 |

moderation by neighborhood deprivation ($\beta_3$), the exogeneity assumption must hold. In other words, the estimates are biased if $Cov(PGS_{ij}, \epsilon_{ij}) = Cov(PGS_{ij} \times GSID_j, \epsilon_{ij}) \neq 0$.

Family data can be leveraged to address this problem as it is possible to decompose the error term: $\epsilon_{ij} = u_j + e_{ij}$, where $u_j$ is an error term varying between DZ twin pairs whereas $e_{ij}$ denotes an idiosyncratic error term, varying within DZ twin pairs. When demeaning the variables in a fixed effects model ($Y_{ij} - \bar{Y}_j = \dot{Y}_{ij}$), $u_j$ drops from the equation, meaning that the exogeneity assumption with respect to $u_j$ can be relaxed. Thus, for the estimate of $PGS_{ij} \times GSID_j$ to be unbiased, one has only to assume exogeneity with respect to $e_{ij}$.

In addition, it is also often argued that in a within-family design the PGS is quasi-exogenous as genetic differences in full biological siblings are randomized at conception as the transmission of alleles from parents to children is a random process [75–77]. So, there are strong reasons to assume PGS as exogenous.

Given that families might select into neighborhood deprivation based on family SES or birth cohort, we additionally control for $PGS_{ij} \times SES_j$ and $PGS_{ij} \times Cohort_j$. FE models draw only on within-family variation. Therefore, the main effects of variables varying only between families (GSID, family SES, and cohort) are not estimated.

The model equation for the fixed effects model is:

$$Y_{ij} = \beta_1 * P\dot{G}S_{ij} + \beta_2 * GSID \times P\dot{G}S_{ij} + \beta_3 * SES \times P\dot{G}S_{ij} + \beta_4 * Cohort \times P\dot{G}S_{ij} + \dot{e}_j$$

We account for the different controls in a stepwise procedure, resulting in four models for the pooled and FE analysis: In the first model (M1), the raw PGS effect is estimated. The second model (M2) adds the interaction between PGS and GISD. The third model (M3) additionally accounts for the interaction between PGS and family SES. The fourth model (M4) additionally controls for the interaction between PGS and birth cohort.

## 4. Results

### 4.1. Pooled analysis

The results of the pooled analysis are shown in Table 4. The first model suggests – on average – a small, but significant ($p < 0.05$) positive genetic effect on life satisfaction on phenotypic life satisfaction. The second model accounts for possible differences in the genetic effect across levels of neighborhood deprivation. M2 shows that for individuals living in neighborhoods with a medium level of neighborhood deprivation, the genetic effect on life satisfaction is about 0.154 ($p < 0.01$), while it is significantly lower for individuals living in less deprived neighborhoods (−0.183, $p < 0.01$). The genetic effect for individuals living in highly deprived neighborhoods is also lower compared to individuals living in mid-deprived neighborhoods. However, the difference is not significant (−0.076, $p > 0.05$). The observed differences in the genetic effect may be

**Table 4. Results of the pooled models (N = 1544).**

|  | M1 | M2 | M3 | M4 |
|---|---|---|---|---|
| PGS | 0.063* | 0.154** | 0.125* | 0.160* |
|  | (0.028) | (0.050) | (0.063) | (0.070) |
| PGS x Low GISD |  | −0.183** | −0.189** | −0.190** |
|  |  | (0.068) | (0.069) | (0.069) |
| PGS x High GISD |  | −0.076 | −0.086 | −0.084 |
|  |  | (0.070) | (0.070) | (0.071) |
| Control for PGS x SES | No | No | Yes | Yes |
| Control for PGS x Cohort | No | No | No | Yes |

+ $p < 0.1$, * $p < 0.05$, ** $p < 0.01$, *** $p < 0.001$.

driven by family SES or twins' birth cohort. Therefore, M3 additionally controls for SES differences in the genetic effect and M4 accounts for cohort differences in the genetic effect. However, this does not substantially affect the estimates of the deprivation differences in the genetic effect on life satisfaction. Fig 1 shows the estimated genetic effects on life satisfaction for the three levels of neighborhood deprivation, suggesting an inverted U-shaped pattern with genetic effects being strongest for individuals living in middle deprived neighborhoods, suggesting the in highly deprived neighborhoods, the realization of a genetic potential is suppressed, while affluent neighborhoods compensate for a genetic risk. While the pooled analysis offers some first insights, the estimates might be biased by unobserved between-family characteristics.

### 4.2. Fixed effects models

Therefore, in a second step, fixed effects models are estimated to account for unobserved between-family characteristic effects (see Table 5). The first fixed effects model shows that the observed positive genetic effect on life satisfaction in the first pooled model attenuates and loses its statistical significance after adjusting for unobserved between-family characteristics. This suggests that unobserved between-family characteristics do confound the PGS-life satisfaction association. M2 shows that, for DZ twins living in neighborhoods with a medium level of neighborhood deprivation, the within-family genetic effect on life satisfaction is about 0.255 ($p < 0.05$), while it is significantly lower for DZ twins living in less deprived neighborhoods (-0.333, $p < 0.05$) and significantly lower for DZ twins living in more deprived neighborhoods (-0.342, $p < 0.05$). Compared to M2, M3 additionally controls for SES differences in the within-family genetic effect, and M4 additionally accounts for both SES and cohort differences in the within-family genetic effect. For DZ twins living in neighborhoods with a medium level of neighborhood deprivation, the within-family genetic effect on life satisfaction increases to 0.404 ($p < 0.01$) in M3, and to 0.542 ($p < 0.001$) in M4. Like the pooled model results, including SES and cohort differences in the within-family genetic effect does not substantially affect the estimates of the deprivation differences in the within-family genetic effect on life satisfaction. Fig 2 shows the estimated within-family genetic effects on life satisfaction for the three levels of neighborhood deprivation, and the estimates are not driven by unobserved between-family characteristics. Similar to the pooled model results, this suggests a robust inverted U-shaped pattern, with genetic effects being strongest for individuals living in moderately deprived neighborhoods. This indicates that in highly deprived neighborhoods, the realization of genetic potential is suppressed, while affluent neighborhoods compensate for genetic risk.

### 5. Discussion

This study examines how genetic influences and neighborhood conditions interact to shape life satisfaction in the German population. Using the TwinLife dataset, we combined a polygenic index (PGI) for subjective well-being with geo-coded data on neighborhood deprivation to analyze G × E. The key contribution of this study lies in extending G × E research to

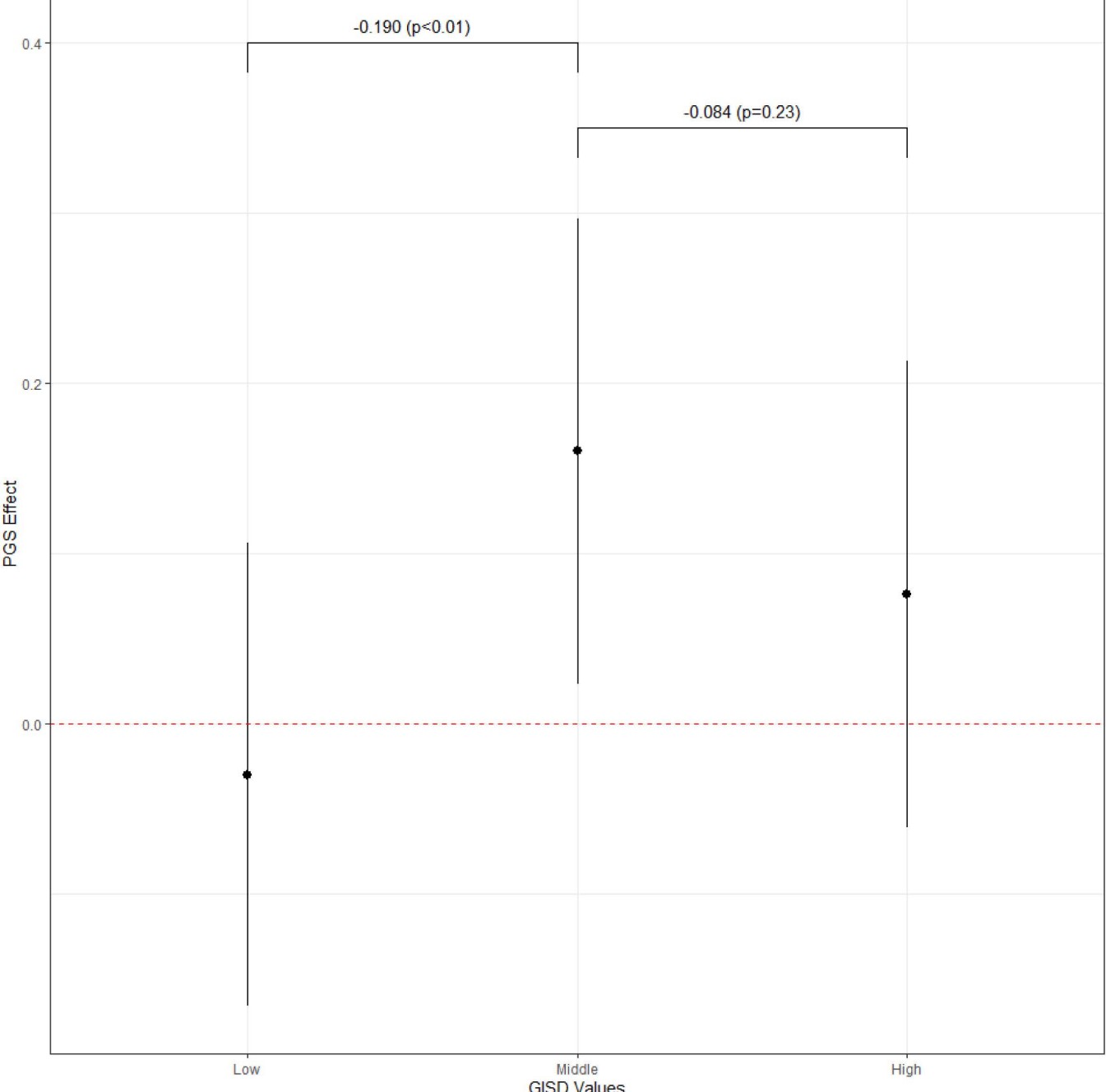

**Fig 1. PGI efffect by GISD groups in the pooled model (M4).**

the domain of life satisfaction—a trait with well-established heritability [13,14]—but limited research on how this genetic influence is moderated by contextual environmental factors such as neighborhood deprivation.

We hypothesized two possible G×E mechanisms: (1) In line with the compensation and triggering models [29], we expected stronger genetic effects in deprived neighborhoods, where disadvantaged environment may trigger genetic

**Table 5. Results of fixed effects models among DZ twins (N = 760).**

|  | M1 | M2 | M3 | M4 |
|---|---|---|---|---|
| PGS (within) | 0.035 | 0.255* | 0.404** | 0.542*** |
|  | (0.062) | (0.105) | (0.135) | (0.148) |
| PGS (within) x Low GISD |  | −0.333* | −0.312* | −0.307+ |
|  |  | (0.157) | (0.158) | (0.158) |
| PGS (within) x High GISD |  | −0.342* | −0.362* | −0.378* |
|  |  | (0.147) | (0.148) | (0.147) |
| Control for PGS x SES | No | No | Yes | Yes |
| Control for PGS x Cohort | No | No | No | Yes |

+ p < 0.1, * p < 0.05, ** p < 0.01, *** p < 0.001.

influences or where wealthy conditions may buffer against them. (2) Alternatively, consistent with the enhancement and suppression models, we hypothesized that advantaged neighborhoods may amplify the effect of positive genetic correlates, while disadvantaged environments might suppress genetic influence.

Our findings reveal a non-linear pattern of genetic influences across levels of neighborhood deprivation. Specifically, the PGI for life satisfaction showed the strongest effect in moderately deprived neighborhoods, with weaker effects observed in both highly deprived and highly advantaged neighborhoods. This pattern partially supports the suppression model: in highly deprived areas, disadvantage environmental constraints may prevent individuals from realizing their genetic potential toward higher life satisfaction. Similarly, the compensation model may explain the dampened genetic influence in advantaged neighborhoods, where positive environmental conditions mitigate genetic disadvantage, reducing variability due to genetic differences.

These results align with theoretical models from neighborhood effects literature. For example, the absence of social capital, institutional resources, and collective efficacy in deprived areas [39,53,57] may function as suppressors, limiting the realization of genetic potential. Conversely, in affluent neighborhoods, high-quality institutions and positive role models may provide compensatory support that narrows the expression of genetic variance [41,49].

Overall, this study makes key contributions. First, it extends G × E research to the domain of subjective well-being, showing that genetic effects on life satisfaction are context dependent. Second, it introduces a nuanced interpretation of neighborhood effects using G × E mechanisms, demonstrating that both suppression and compensation processes may operate simultaneously in different environmental contexts. Notably, our co-twin control design strengthens the robustness of our findings by accounting for unobserved genetic and family-level confounders, while the use of the GSID provides a precise and contextually grounded measure of neighborhood deprivation

While we highlight strengthens of our study, several limitations must be acknowledged. The study is restricted to the German context and may not generalize across countries with different neighborhood dynamics. This is an observational association study, and we do not make any claims about causal relationships. Its cross-sectional nature further limits the ability to infer developmental trajectories of G × E interactions. While PGIs are valuable predictors, they capture only a fraction of total heritability and reflect not only biological influences but also social, demographic, and environmental structures [78]. Future research should employ longitudinal data to track changes over time and examine specific neighborhood mechanisms, such as peer effects, social control, and access to institutional resources, in greater detail. Additionally, incorporating diverse samples across different national contexts would help clarify the extent to which these G × E patterns are structurally or culturally specific.

## 6. Implication and recommendation

This study offers important implications for both theory and practice by highlighting how genetic influences on life satisfaction are contingent upon environmental context. Theoretically, our findings contribute to the GXE interaction literature by

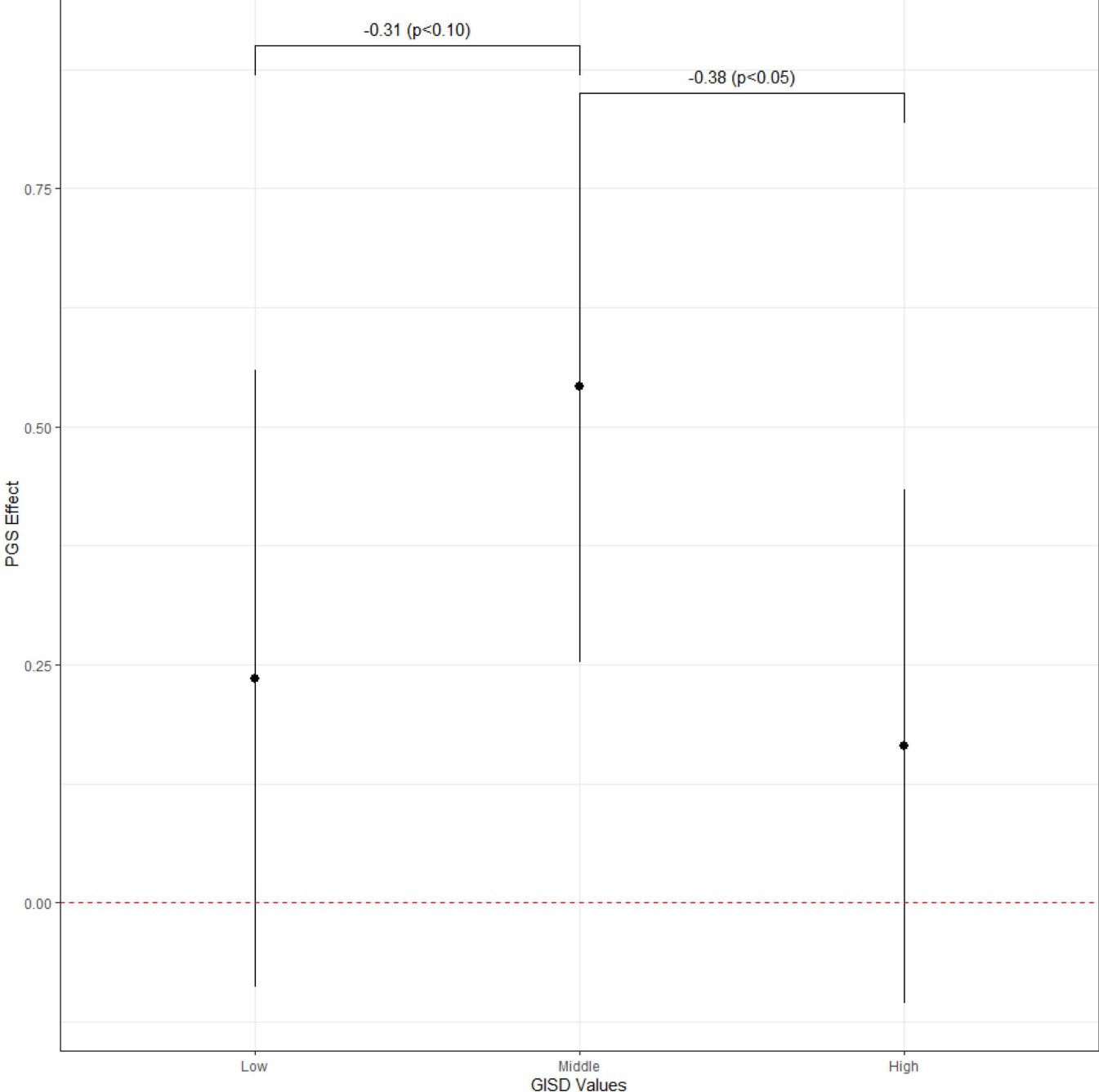

**Fig 2. PGI effect by GISD groups in the fixed-effects model (M4).**

demonstrating that neighborhood conditions, such as levels of deprivation, moderate the expression of genetic correlates of well-being. This advances G × E frameworks [29] by providing empirical support for both suppression and compensation mechanisms, thereby illustrating how environmental disadvantage can constrain or buffer genetic influences. Such insights contribute to psychology by emphasizing that individual differences in well-being cannot be fully understood

without accounting for structural (i.e., socio-economic status) [79] and ecological variables (i.e., greener space) [80]. From a genetic research perspective, our results underscore that polygenic indices are context-sensitive rather than universally predictive, reinforcing the importance of embedding PGIs within socio-environmental frameworks. Practically, the findings suggest that improving neighborhood conditions, such as enhancing access to social support, institutional resources, and collective efficacy, could strengthen individual well-being by enabling the realization of genetic potential. These results point toward the value of interdisciplinary approaches that bridge genetics, psychology, and social policy to more effectively address disparities in mental health and life satisfaction.

## 7. Conclusion

This study contributes to the growing literature on the determinants of life satisfaction by highlighting the significant roles of both genetic influences and neighborhood conditions. Our findings underscore the importance of considering gene-environment interactions when studying life satisfaction and suggest the genetic influences can vary significantly depending on the environmental context. By advancing our understanding of these interactions, we can better inform policies and interventions aimed at enhancing life satisfaction and overall well-being.

AcknowledgmentsWe would like to thank Charlotte Pahnke, Andreas Forstner, Markus Nöthen, Carlo Maj, and Shirin Zare for their contributions to DNA extraction and Polygenic index calculations. We also extend our appreciation to Anita Kottwitz from the TwinLife data management team for her invaluable support. Additionally, we are grateful for the insightful feedback provided by colleagues during the TwinSNPs and TECS organizational meetings.

## Author contributions

**Conceptualization:** Nadia Victoria Harerimana, Mirko Ruks.

**Data curation:** Mirko Ruks.

**Formal analysis:** Nadia Victoria Harerimana, Yixuan Liu, Mirko Ruks.

**Visualization:** Nadia Victoria Harerimana, Mirko Ruks.

**Writing – original draft:** Nadia Victoria Harerimana, Yixuan Liu, Mirko Ruks.

**Writing – review & editing:** Nadia Victoria Harerimana, Yixuan Liu, Mirko Ruks.

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
