## [Decision Letter · Decision Letter 0]

21 Apr 2025

Dear Dr. Liu,

Thank you for submitting your manuscript to PLOS ONE. After careful consideration, we feel that it has merit but does not fully meet PLOS ONE’s publication criteria as it currently stands. Therefore, we invite you to submit a revised version of the manuscript that addresses the points raised during the review process.

**ACADEMIC EDITOR:**

We look forward to receiving your revised manuscript.

Kind regards,

Hoh Boon-Peng, PhD

Academic Editor

PLOS ONE

Journal Requirements:

“The TwinLife study was funded by the German Research Foundation (Grant No. 220286500), with funding awarded to Martin Diewald, Christian Kandler, Frank M. Spinath, Bastian Mönkediek, and Rainer Riemann. The molecular genetic extension of TwinLife was also funded by the German Research Foundation (Grant No. 428902522), with funding awarded to Martin Diewald, Peter Krawitz, Markus M. Nöthen, Rainer Riemann, and Frank M. Spinath. The funders had no role in the study design, data collection and analysis, decision to publish, or manuscript preparation. YL and MR are supported by the TwinLife project, while NVH is funded by the European Union’s HORIZON-MSCA-2021-DN-01 programme (Grant No. 101073237).

“Please provide an amended statement that declares *all* the funding or sources of support (whether external or internal to your organization) received during this study, as detailed online in our guide for authors at http://journals.plos.org/plosone/s/submit-now.  Please also include the statement “There was no additional external funding received for this study.” in your updated Funding Statement.

“, Rainer Riemann, and Frank M. Spinath. The funders had no role in the study design, data collection and analysis, decision to publish, or manuscript preparation. YL and MR are supported by the TwinLife project, while NVH was funded by the European Union, under project #101073237 – European Society of Social Genetics Network. We would like to thank Charlotte Pahnke, Andreas Forstner, Markus Nöthen, Carlo Maj, and Shirin Zare for their contributions to DNA extraction and polygenic score calculations”

“The TwinLife study was funded by the German Research Foundation (Grant No. 220286500), with funding awarded to Martin Diewald, Christian Kandler, Frank M. Spinath, Bastian Mönkediek, and Rainer Riemann. The molecular genetic extension of TwinLife was also funded by the German Research Foundation (Grant No. 428902522), with funding awarded to Martin Diewald, Peter Krawitz, Markus M. Nöthen, Rainer Riemann, and Frank M. Spinath. The funders had no role in the study design, data collection and analysis, decision to publish, or manuscript preparation. YL and MR are supported by the TwinLife project, while NVH is funded by the European Union’s HORIZON-MSCA-2021-DN-01 programme (Grant No. 101073237).”

Reviewers' comments:

Reviewer's Responses to Questions

**Comments to the Author**

1. Is the manuscript technically sound, and do the data support the conclusions?

Reviewer #1: Yes

2. Has the statistical analysis been performed appropriately and rigorously?

Reviewer #1: Yes

3. Have the authors made all data underlying the findings in their manuscript fully available?

Reviewer #1: Yes

4. Is the manuscript presented in an intelligible fashion and written in standard English?

Reviewer #1: Yes

Reviewer #1: PONE-D-24-57036

Neighbourhood Deprivation, Genetic Predisposition, and Life Satisfaction: Evidence from the German Twin Family Panel

Interesting and excellent study.

Overall, the study is important and contributes to the genetic and psychology research area.

There are some suggestions and comments as below:

(1) The study is focusing on gene-environment interactions (GxE) on individual life satisfaction. Participants between

10 and 24 years old. There is a big gap in individual developmental stages.

From a psychological view, it involved the development from early adolescence to young adults. Many other

factors (including developmental tasks) will influence individual life satisfaction. Kindly justify.

(2) Life Satisfaction—Instrument

a) How many total items were used for data collection?

b) What do you mean by “comprising five items each”?

c) How does CFA use to measure satisfaction level? Any data for CFA? The below statement is a bit confusing.

Kindly revise and justify.

“The final life satisfaction score was created using confirmatory factor analysis of the raw indicators” .

(3) Methodology – Data Collection.

a) Kindly justify when the data collection is done. The below statements are not reported deeply.

“The data were accessed for research purposes on May 2024.”

“Data collection for the Face-to-Face (F2F 1) survey occurred in two phases: Subsample A was collected

between September 28, 2014, and May 28, 2015, while Subsample B was collected between September

16, 2015, and April 18, 2016”

(4) Family SES is measured as a factor based on parental years of education, occupational status (ISEI), and OECD

household net income. However, it is not reported clearly in the main document. Kindly justify and report the data

for the below statements:

“ .. in the statistical analysis we additionally control for family SES, measured as a factor based on parental

years of education, occupational status (ISEI) and OECD household net income. Additionally, we control for

birth cohort status, using a categorical variable indicating whether a twin belongs to cohort 2, 3, or 4.”

(5) Discussion lacks citation or previous study to support. Suggest discussing the findings through theories as

mentioned in the subheading “Theoretical Framework.”. Also, discuss the main finding and overall contribution.

(6) Suggest to add the subheading “Implication and Recommendation” – Explain the implication from a theoretical

and practical view. How does this study contribute to both the genetic and psychology areas?

**Do you want your identity to be public for this peer review?** For information about this choice, including consent withdrawal, please see our Privacy Policy

Reviewer #1: No

---

## [Author Response · Author response to Decision Letter 1]

23 Jul 2025

Comment 1:

The study is focusing on gene-environment (GxE) on individual life satisfaction. Participants between 10 and 24 years old. There is a big gap in individual developmental stages. From a psychological view, it involved the development from early adolescence to young adults. Many other factors (including developmental tasks) will influence individual life satisfaction. Kindly justify.

Response:

Thank you very much for highlighting the importance of age-related differences in our analysis. We fully acknowledge that our sample spans a broad developmental range—from early adolescence to young adulthood—which may introduce variability in the GxE estimates due to age or cohort effects.

While we recognize the value of investigating these developmental factors, modeling a three-way interaction (PGS × Neighborhood × Age) would significantly increase the complexity of our analysis and likely exceed our available statistical power to yield robust estimates. For this reason, we have focused our current analysis on the overall gene-environment interaction (GxE), averaged across the age range.

We agree that exploring potential age-specific patterns in GxE interactions is an important avenue for future research and appreciate you drawing attention to this point.

Comment 2

Life Satisfaction—Instrument

Response:

We appreciate the Reviewer’s comment and have added more description about the instrument the Life satisfaction subheading. See below:

Participants were asked, “How satisfied are you with your life in general?” and responded to a set of five statements reflecting their cognitive evaluation of overall life satisfaction using a five-point Likert scale, ranging from 1 (“Disagree strongly”) to 5 (“Agree strongly”). For individuals aged 16 years and older, the items included: (1) “My life is close to ideal,” (2) “The conditions of my life are excellent,” (3) “I am satisfied with my life,” (4) “So far, I have gotten the important things I want in life,” and (5) “If I could live my life over, I would change almost nothing.” For participants aged 10 to 15 years, the items were as follows: (1) “My life is going well,” (2) “I have a good life,” (3) “I am happy with my life,” (4) “I have gotten the things I want in life,” and (5) “If I could live my life over, I would want the same life.” The final life satisfaction score was derived separately for each version through confirmatory factor analysis (CFA) of the raw indicators, allowing for the estimation of a latent construct that captures the underlying structure of life satisfaction. For participants aged 16 years and older, the model fit was: *χ*^2^ (5) =17.10, *χ* = .004, RMSEA = .054, CFI = .992, TLI = .984, SRMR = .016. The reliability indices were: Cronbach’s *𝛼* = .84, and McDonald’s *ω* = .85. For participants aged 10 to 15 years, the model fit was: *χ*^2^ (5) = 26.36, *χ* < .001, RMSEA = .077, CFI = .988, TLI = .976, SRMR = .018. The reliability indices were: Cronbach’s *𝛼* = .86, McDonald’s *ω* = .87.

Comment 3

Methodology – Data Collection.

Kindly justify when the data collection is done. The below statements are not reported deeply.

Response:

We appreciate the Reviewer’s comment and have added a clarifying sentence under the Participants subheading to specify the timing and scope of data collection. See below:

“In this study, we included participants from Cohorts 2 (twins born in 2003/2004), 3 (twins born in 1997/1998), and 4 (twins born between 1990 and 1993) who participated in both the first (F2F1) and third (F2F3) face-to-face waves of the TwinLife study and who provided saliva samples during F2F3. The F2F1 wave was conducted between September 2014 and May 2015 (Subsample A), and between September 2015 and April 2016 (Subsample B). The F2F3 wave took place from November 2018 to July 2019 (Subsample A), and from September 2019 to June 2020 (Subsample B).”

Comment 4:

Family SES is measured as a factor based on parental years of education, occupational status (ISEI), and OECD household net income. However, it is not reported clearly in the main document. Kindly justify and report the data for the below statements:

Response:

We thank the Reviewer for this comment. In response, we have added a description of the confirmatory factor analysis model fit and reliability indices for Family SES in the Covariates subheading.

“The CFA model for family SES was saturated, yielding perfect fit indices: *χ*^2^ (0) = 0, *χ* = NA, RMSEA = 0, CFI = 1.00, TLI = 1.00, SRMR = 0. The reliability indices were: Cronbach’s *𝛼* = .40 and McDonald’s $= .70. Although Cronbach’s *𝛼* was relatively low—likely due to violations of the tau-equivalence assumption, McDonald’s *ω* indicated acceptable construct reliability. Given that *ω* does not assume equal factor loadings and is more appropriate in structural equation modeling contexts, we consider the construct to demonstrate adequate internal consistency.”

Comment 5:

Discussion lacks citation or previous study to support. Suggest discussing the findings through theories as mentioned in the subheading “Theoretical Framework.”. Also, discuss the main finding and overall contribution.

Response:

We thank the Reviewer for this insightful comment. In response, we have substantially revised the Discussion section to more clearly engage with the theoretical framework outlined earlier in the manuscript. Specifically, we now interpret our findings through the lens of established G×E mechanisms (e.g., compensation, suppression) and neighborhood effects theories (e.g., collective efficacy, institutional resources), supported by relevant literature (e.g., Shanahan & Hofer, 2005; Sampson et al., 1997; Ellen & Turner, 1997). We also explicitly highlight the main findings and articulate the overall contribution of the study—namely, extending G×E research to the domain of life satisfaction and contextualizing genetic influences within varying neighborhood conditions. These revisions are reflected in the revised Discussion section of the manuscript.

Comment 6:

Suggest to add the subheading “Implication and Recommendation” – Explain the implication from a theoretical and practical view. How does this study contribute to both the genetic and psychology areas?

Response:

We appreciate the Reviewer’s suggestion to include a dedicated subheading on Implication and Recommendation. In response, we have added a new section that outlines both the theoretical and practical implications of our findings. This section highlights the study’s interdisciplinary contribution—bridging behavioral genetics and psychological research on subjective well-being—while also emphasizing how neighborhood-level interventions may moderate genetic risks or support genetic potentials. The new section is titled Implication and Recommendation and can be found following the Discussion.

---

## [Decision Letter · Decision Letter 1]

31 Jul 2025

Neighborhood Context, Genetic influences, and Life Satisfaction: Evidence from the German Twin Family Panel

PONE-D-24-57036R1

Dear Dr. Harerimana,

We’re pleased to inform you that your manuscript has been judged scientifically suitable for publication and will be formally accepted for publication once it meets all outstanding technical requirements.

Kind regards,

Hoh Boon-Peng, PhD

Academic Editor

PLOS ONE

Additional Editor Comments (optional):

The comments have been addressed accordingly, hence recommend accept for publication

Reviewers' comments:

Reviewer's Responses to Questions

**Comments to the Author**

Reviewer #1: All comments have been addressed

2. Is the manuscript technically sound, and do the data support the conclusions?

Reviewer #1: Yes

3. Has the statistical analysis been performed appropriately and rigorously?

Reviewer #1: Yes

4. Have the authors made all data underlying the findings in their manuscript fully available?

Reviewer #1: Yes

5. Is the manuscript presented in an intelligible fashion and written in standard English?

Reviewer #1: Yes

Reviewer #1: Accept with no further comments.

Thank you.

**Do you want your identity to be public for this peer review?** For information about this choice, including consent withdrawal, please see our Privacy Policy

Reviewer #1: No

---

## [Editor Report · Acceptance letter]

PONE-D-24-57036R1

PLOS ONE

Dear Dr. Harerimana,

I'm pleased to inform you that your manuscript has been deemed suitable for publication in PLOS ONE. Congratulations! Your manuscript is now being handed over to our production team.

Kind regards,

on behalf of

Professor Dr Hoh Boon-Peng

Academic Editor

PLOS ONE